# Predicting Cardiovascular Disease Mortality: Leveraging Machine Learning for Comprehensive Assessment of Health and Nutrition Variables

**DOI:** 10.3390/nu15183937

**Published:** 2023-09-11

**Authors:** Agustin Martin-Morales, Masaki Yamamoto, Mai Inoue, Thien Vu, Research Dawadi, Michihiro Araki

**Affiliations:** 1Artificial Intelligence Center for Health and Biomedical Research, National Institutes of Biomedical Innovation, Health and Nutrition, 3-17 Senrioka-shinmachi, Settsu 566-0002, Japan; 2National Cerebral and Cardiovascular Center, 6-1 Kishibe-Shinmachi, Suita 564-8565, Japan; 3Graduate School of Medicine, Kyoto University, 54 Shogoin-Kawahara-cho, Sakyo-ku, Kyoto 606-8507, Japan; 4Graduate School of Science, Technology and Innovation, Kobe University, 1-1 Rokkodai, Nada-ku, Kobe 657-8501, Japan

**Keywords:** machine learning, cardiovascular disease, prediction model, nutrition, dietary features, SHAP

## Abstract

Cardiovascular disease (CVD) is one of the primary causes of death around the world. This study aimed to identify risk factors associated with CVD mortality using data from the National Health and Nutrition Examination Survey (NHANES). We created three models focusing on dietary data, non-diet-related health data, and a combination of both. Machine learning (ML) models, particularly the random forest algorithm, demonstrated robust consistency across health, nutrition, and mixed categories in predicting death from CVD. Shapley additive explanation (SHAP) values showed age, systolic blood pressure, and several other health factors as crucial variables, while fiber, calcium, and vitamin E, among others, were significant nutritional variables. Our research emphasizes the importance of comprehensive health evaluation and dietary intake in predicting CVD mortality. The inclusion of nutrition variables improved the performance of our models, underscoring the utility of dietary intake in ML-based data analysis. Further investigation using large datasets with recurring dietary recalls is necessary to enhance the effectiveness and interpretability of such models.

## 1. Introduction

### 1.1. Background

Cardiovascular disease (CVD) is a dominant global health concern, persistently ranking as one of the primary causes of death. This broad classification of the diseases affecting the heart and blood vessels includes conditions such as coronary artery disease, heart failure, and stroke. These health problems claim millions of lives each year, emphasizing the urgency of developing effective preventive strategies [1].

Increasingly, researchers are recognizing the potential of nutritional health to dramatically influence CVD mortality rates [2,3,4,5]. Nutritional health pertains to the intake of the essential nutrients that are necessary for bodily functions and overall well-being. From consuming the correct caloric amounts, achieving a balanced intake of macronutrients (such as proteins, fats, and carbohydrates), to ensuring adequate levels of micronutrients like vitamins and minerals, nutrition can influence various physiological functions, including cardiovascular health. However, despite its importance, a comprehensive integration of these multifaceted nutritional factors into existing cardiovascular risk prediction models remains a challenge.

Current risk models often struggle to accurately encapsulate an individual’s nutritional profile. The broad spectrum of nutritional metrics is complex, presenting obstacles to clinicians and researchers attempting to synthesize this information into meaningful data [6,7]. However, as lifestyle factors—primarily unhealthy dietary habits—are significantly contributing to the rising prevalence of CVD, it is imperative to effectively incorporate these factors into risk models, enabling more accurate and personalized preventive strategies.

The emergence of machine learning (ML) offers the potential to overcome some of the limitations of current predictive models. ML allows for the handling of the intricate array of nutritional data and their nonlinear relationships in ways that traditional regression models cannot [6,7]. These advanced techniques can process vast and sparse data matrices, identifying complex patterns and relationships among a multitude of variables. However, ML models are not without their problems: their ‘black-box’ nature makes them difficult to interpret. While feature importance-based explanations can provide some insight into ML models’ inner workings, it remains a challenge to manage issues such as multicollinearity and redundant variables.

Fortunately, comprehensive databases such as the National Health and Nutrition Examination Survey (NHANES) allow for detailed nutritional profiling. By employing advanced ML techniques on such extensive data sets, it becomes feasible to discern complex interactions between different types of nutrients and their nonlinear relationships with CVD outcomes—far beyond the scope of traditional regression models or composite nutritional indices [8,9,10,11]. However, there’s still uncertainty regarding whether nutritional data from dietary recalls can significantly enhance CVD mortality risk prediction beyond that achieved using traditionally used biomarkers, such as blood pressure or cholesterol levels.

In our pursuit to optimize CVD risk prediction, it is crucial that we consider advanced analytical methods and integrate detailed dietary data into these models. The ultimate goal is to enhance the precision of risk estimations, allowing for more effective preventive measures and early intervention strategies. This progress will not only increase the effectiveness of public health efforts but also provide individuals with actionable insights to manage their cardiovascular health.

### 1.2. Overview

In this study, we strove to formulate ML-based prediction models for CVD fatalities. We employed three distinct models, namely, health, nutrition, and a mixed model that encapsulated both aspects. These models were designed to streamline variables into several ML algorithms, negating multicollinearity and redundancy. Our goal was to improve the integrity of the data, while maintaining the interpretability of the models.

To assess the efficacy of each ML algorithm, we employed various performance metrics such as the area under the curve (AUC), accuracy (ACC), recall, precision, and F1 score. In terms of feature selection, we uses variance inflation factor (VIF)-based filtering, which efficiently expels multicollinearity and redundant variables. This mechanism aids in preserving the comprehensibility of a model. Additionally, we also implemented class-balancing strategies, aiming to enhance the confidence score of the classification models.

Interpreting the model entailed identifying the salient risk factors for CVD. For this purpose, we employed Shapley additive explanation (SHAP) values, a theoretically grounded post hoc model-interpretation technique. SHAP values allowed us to discern the critical risk factors for CVD and understand the nature of the relationship between these factors and CVD.

## 2. Materials and Methods

### 2.1. Study Design

The research material consisted of data gathered from the NHANES from the years 2003 through 2012. NHANES, a major program of the National Center for Health Statistics (NCHS), provides critical information on the health behaviors, prevalence of chronic diseases, and nutritional status of individuals in the United States. though biennial surveys. This complex, multistage probability cluster survey utilizes sampling methods designed to represent the general U.S. population, involving aspects such as stratification and unequal sampling probabilities.

The nature of the NHANES design introduces potential for nonparticipation errors, nonresponse errors, and sample design errors. To mitigate these biases, NHANES provides complex sampling weights, allowing researchers to adjust their data analyses appropriately. In our research, we adopted these sampling weights into ML-model-based analysis, a deviation from the conventional logistic regression-based methods.

The data we used were formed from an extensive sample of 50,912 participants with provided health and dietary questionnaire responses and health examination variables. In line with the NHANES data analysis guideline, we integrated data from 2003 to 2012. We applied stringent exclusion criteria, removing participants who were under 20 years of age, had a previous history of CVD, provided incomplete dietary data, and reported daily caloric intake below 800 kcal or above 6000 kcal. Consequently, we selected a final study population of 9706 participants.

After undersampling, the dataset was randomly partitioned into a training set (*n* = 345 (80%)) and a test set (*n* = 87 (20%)). The flow diagram detailing our data preparation process is presented in Figure 1.

Our study was conducted using Python 3.7.11 (https://www.python.org/, accessed on 1 June 2023) along with compatible open-source packages for data analysis and ML model building.

### 2.2. Primary Outcome

The primary outcome of this research centered on the prediction of CVD-related mortality leveraging ML algorithms with NHANES data from 2003 to 2012. The occurrence of CVD-related mortality was ascertained via National Death Index (NDI) linkage, specifically, deaths due to heart disease or cerebrovascular diseases. CVD mortality was defined as time from NHANES interview to the time of CVD-related death. In cases where participants died from other causes, these were treated as censored data. Our inclusion criteria consisted of participants aged over 20 years at the time of the interview who had no prior history of CVD. No measures were implemented to blind assessment of predictors for the outcome or other predictors. Furthermore, no measures were taken to blind the assessment of the outcome itself.

### 2.3. Data Preprocessing and Variable Selection

From NHANES 2003–2012, we extracted the health questionnaire and examination variables that were collected every two years. Any variables with more than 30% missing values were not considered for analysis. From the remaining data, we selected 33 variables for the nutrition model; these variables were extracted from the total nutrient intakes files except for the “Milk_int” variable, which was created by combining several variables from the diet and behavior questionnaires. The health model comprises all the non-nutrition-related variables collected via demographic, laboratory, examination, and questionnaire methods. More information about the selected variables can be found in Appendix A. Missing data under 30% were imputed using mode replacement for categorial variables and Scikit-Learn IterativeImputer (RandomForestRegressor model) for numerical variables.

Variables to be removed were determined by considering the clinical meaning of the variables, VIF, and correlation coefficients between variables. We removed multicollinearity to help distinguish how much each variable influenced the regression or classification. Multicollinearity was calculated using the VIF of each variable and correlation coefficients among variables. For this study, a cutoff VIF value of less than 10 was selected. A variable was considered for removal if its VIF value exceeded this limit. The final set of predictors considered for our ML models included demographic variables such as age, sex, race (black, Hispanic), covered by health insurance, and education level. CVD risk factors that were taken into account included body mass index (BMI, kg/m^2^), waist circumference (cm), total cholesterol (mg/dL), high-density lipoprotein (HDL) and low-density lipoprotein (LDL) cholesterol levels (mg/dL), hemoglobin (g/dL), glycated hemoglobin (%), systolic and dyastolic blood pressure (mm Hg), diabetes status (yes/no), alcohol drinking status (nondrinker/drinker), and current smoking status (smoking/quit/never). Other relevant blood biomarkers considered were calcium (mg/dL), iron (mcg/dL), potassium (mEq/L), sodium (mg/dL), phosporus (mmol/L), and uric acid (mg/dL). The estimated glomerular filtration rate eGFR was calculated from creatinine values (mg/dL) according to the following CKD-EPI equation:(1)$$eGFR=141\times \min{\left(\frac{S_{cr}}{\kappa },1\right)}^{\alpha }\times \max{\left(\frac{S_{cr}}{\kappa },1\right)}^{-1.209}\times 0.993^{\mathrm{Age}}\times 1.018\;\left[\mathrm{if}\;\mathrm{female}\right]\times 1.159\;\left[\mathrm{if}\;\mathrm{black}\right]$$
where $S_{cr}$ is serum creatinine in mg/dL, κ is 0.7 for women and 0.9 for men, α is −0.329 for women and −0.411 for men, min indicates the minimum of $S_{cr}$/κ or 1, and max indicates the maximum of $S_{cr}$/κ or 1.

Furthermore, nutrition variables were considered, including daily standardized intake of micronutrients (e.g., sodium, magnesium) and macronutrients (e.g., saturated fat, sugar, protein). In addition to statistical considerations, the final selection of nutritional features was influenced by their established relevance in the literature concerning CVD mortality. For instance, total sugar intake, fiber intake, and total carbohydrate intake presented high collinearity. We decided to remove total carbohydrate intake based on the greater relevance of the other two variables to CVD and other metabolic-syndrome-related diseases. These dietary elements were averaged from the total nutrient intake files of two independent 24 h dietary recalls conducted immediately after the NHANES interview.

### 2.4. Training of Machine Learning Models

Once the variables were chosen, we set five ML models for the purpose of death from CVD prediction: logistic regression, SVM, RF, XGBoost, and LigthGBM. We utilized a 5-fold cross-validation approach. The cross-validation method is a proficient way to safeguard against overfitting by testing the model with various combinations of training and validation data.

The class proportion within the study group was significantly skewed (CVD:non-CVD = 1:43.9). The model, in the case of such an imbalanced dataset, could overfit the non-CVD instances, leading to low sensitivity performance. Thus, we adjusted the ratio between CVD and non-CVD cases to achieve balance, employing undersampling, ensuring the model was adequately trained on CVD patient cases. For the implementation of model training, we utilized Scikit-Learn version 1.0.2 (https://scikit-learn.org, accessed on 5 April 2022).

### 2.5. Performance Metrics

The performance of the different models was measured and compared to assess their predictive capabilities for the different outcomes—health, nutrition, and mixed. The evaluation of the binary models (cases versus noncases) was conducted using several performance metrics. Balanced accuracy, denoted as ACC, is the average of sensitivity and specificity, providing a more balanced measure in the presence of unequal class distribution. Sensitivity, also known as recall, represents the proportion of actual positive cases (TP) correctly identified as such from the total positive cases, i.e., TP/(TP + FN). Specificity, on the other hand, is the proportion of actual negative cases (TN) correctly identified from the total negatives, i.e., TN/(TN + FP). Precision, also known as the positive predictive value, is the proportion of true positive cases from all cases predicted as positive, i.e., TP/(TP + FP). This metric provided insight into the correctness achieved by our model when it predicted a case as positive. The AUC is a measure of a model’s ability to distinguish among classes. It provides an aggregate measure of performance across all possible classification thresholds. The F1 score is the harmonic mean of precision and recall. It is a balanced measure for binary classification models, especially useful when the class distribution is uneven. It can be calculated as:(2)$$F1=\frac{2\times \mathrm{precision}\times \mathrm{recall}}{\mathrm{precision}+\mathrm{recall}}$$

### 2.6. Variable Importance

ML models are often viewed as inscrutable due to the complexity of understanding how these algorithms yield precise forecasts for specific patient groups. To mitigate this issue, we used the SHAP values in our research, a comprehensive framework originally proposed by Lundberg and Lee [12]. By utilizing SHAP, we were able to deliver reliable and locally precise feature attributions for our random forest predictive models, intended to identify the primary determinants of mortality from CVD. SHAP values are used to evaluate the contribution of each feature to an outcome when included in the model, while considering the potential interactions with all other features. This method enabled us to justify the rationale behind our predictive model, which encapsulates pertinent risk factors contributing to mortality. It also allowed us to assess the significance of the feature rankings within our final models.

## 3. Results

### 3.1. Variable Selection and General Characteristics

Our study began with the extraction of all variables that had potential relevance to CVD. These were extracted from the NHANES database. Following the filtering described in the Section 2, we were left with 59 selected variables that were utilized in the development of our CVD classifiers. To ensure representative outcomes, complex sampling weights were applied to define our study population, which minimized bias in the original dataset.

The study population comprised 9706 participants. Of this number, 216 (2.3%) had CVD. Table 1 shows the general characteristics of the variables used in this study with a *p*-value lower than 0.001. Those with CVD were predominantly older, with a median age of 68 years compared with 43 years in the control group; were more likely to be male; and had higher BMI, mean waist size, systolic blood pressure, HDL, glycated hemoglobing, hemoglobin, potassium, lactate dehydrogenase, sodium, and uric acid levels. They also had a higher rate of health insurance coverage, were less educated (with most not having achieved education beyond ninth grade), and had a higher rate of smoking or being former smokers.

We also observed differences in dietary intakes between those with CVD and those without (bottom part in Table 1). Those with CVD had lower median intakes of calcium, copper, fiber, folate, magnesium, niacin, polyunsaturated fatty acids, protein, sodium, sugar, and vitamins B1, B6, and E.

The data corresponding to the full list of variables including those with *p* value above 0.001 can be found in Appendix A.

### 3.2. Comparison of Model Performance

Table 2 presents the performance of various approaches for the three different models: health, nutrition, and mixed. The table includes ACC, AUC, recall, precision, and F1 score for each algorithm under the three models. The highest performance for each model is highlighted in bold. In the health category, it is notable that the random forest and ThunderSVM models demonstrated superior performance in predicting death from CVD. Specifically, the random forest model was particularly proficient, with an accuracy and all other metrics (recall, precision, and F1 score) standing at 0.8. Meanwhile, ThunderSVM followed closely, showing a balanced performance across all metrics, with a value of 0.79, and outperforming random forest with an AUC of 0.88. Other models, namely logistic regression, XGBoost, and LightGBM, exhibited reasonable but slightly lower performance metrics. Within the nutrition category, random forest was again the algorithm with the best performance overall. This algorithm displayed superior performance with accuracy, AUC, recall, precision, and F1 score all evenly balanced at 0.7. Other models, including logistic regression and LightGBM, also showed noteworthy results, with a highest AUC value of 0.7. However, they fell short in comparison to the random forest model’s overall consistency. In the mixed category, incorporating both health and nutrition data, we observed variations in the performance of different models. Notably, the random forest model emerged as the top performer, achieving accuracy, recall, precision, and F1 score of 0.82 and an AUC of 0.88. Similarly, the XGBoost and LightGBM models exhibited competitive performance, with all their metrics but AUC hovering around the 0.8 and 0.79 marks, respectively, and an AUC of 0.87. The results demonstrated varying degrees of effectiveness across different models in predicting death from CVD, based on the considered data categories. In the health category, the random forest and ThunderSVM models exhibited close performance metrics, although the random forest model slightly outperformed the others. In the nutrition category, all models generally performed less effectively than in the health category, with the random forest model once again proving to be the superior algorithm. For the mixed category, despite all models showing improved metrics, the random forest model continued to consistently outperform the other models across all metrics.

### 3.3. Predictive Variable Analysis

SHAP values delineate the relative importance of each variable to the predicted outcome within individual instances. We aggregated the SHAP values over the test set to distinguish the top influencers for each of the three models: health, nutrition, and mixed. Variables were ranked in descending order based on their average SHAP values across these models.

As indicated by the SHAP values in Figure 2, age was found to be the most potent variable overall, with SHAP values of 0.13 and 0.1 for the health and mixed models, respectively. For the health model, systolic blood pressure (0.03) and several other factors like uric acid, hemoglobin, HbA1C, lactic acid, and waist circumference also demonstrated a positive correlation, signifying that higher values of these variables tend to classify the individual into a higher health risk category. On the other hand, eGFR and diastolic blood pressure showed a negative correlation, indicating that lower values of these are associated with increased risk.

In the nutrition model, fiber (0.05) and calcium (0.03) were the most influential variables that were negatively correlated with the risk of death from CVD, followed by polyunsaturated fatty acids, vitamin E and magnesium (0.02), and vitamin B6 and protein (0.01), suggesting that lower levels of these nutrients in the diet are linked to a higher risk of death from CVD. Interestingly, vitamin B2 (0.03), potassium (0.02), and sodium (0.01) demonstrated a positive correlation with the risk of death from CVD.

The mixed model, combining health and nutrition variables, had influential variables similar to those of the individual models, with age and eGFR showing high importance and correlations being consistent with those observed in the individual models.

It is important to note that the interpretation of SHAP values depends on the directionality of the variable’s relationship with the predicted outcome. As shown in the bottom part in Figure 2, positive and negative SHAP values imply an increased or decreased likelihood of a higher risk of death from CVD, respectively.

The potential for class imbalance to lower model output and consequently reduce SHAP values was taken into account. Also, feature selection was performed to prevent multicollinearity in the data and ensure accurate importance estimation for crucial variables such as age, systolic blood pressure, and eGFR. The full list of SHAP values for all variables is presented in Appendix A for the mixed model, Appendix A for the health model, and Appendix A for the nutrition model.

## 4. Discussion

The ever-growing incidence of CVD can be largely attributed to the adaptation of Western eating habits and health behaviors, consequently leading to rises in related complications and mortality rates [2,13,14]. Due to the high socioeconomic burden associated with chronic diseases, there is a growing policy interest in the proactive identification and management of risk factors, even before the onset of disease. To address this, we utilized data from the NHANES conducted between 2003 and 2012. We aimed to identify risk factors associated with CVD mortality, considering demographics, diet, and lifestyle, as well as physiological aspects.

Examining the general characteristics of our study population, we observed that individuals diagnosed with CVD were predominantly older, male, had a higher BMI, and showed elevated levels of several clinical parameters, such as systolic blood pressure, HDL, and uric acid. These observations largely align with the known demographic and physiological risk factors of CVD [8,9,10]. Glycated hemoglobin, uric acid, and eGFR, which are markers of diabetes, gout, and kidney disease, respectively, have previously been associated with CVD [15,16,17]. The higher prevalence of smoking and lower education levels observed among those who died from CVD have also been previously reported [9,10].

Participants who died from CVD exhibited significantly lower median intakes of multiple nutrients. Several of the nutrients, such as polyunsaturated fats, magnesium, and dietary fiber, have been previously implicated in cardiovascular health due to their roles in modulating cholesterol levels, inflammation, or overall cardiac function [18,19,20,21]. Rigdon and Basu [10] also identified a protective correlation between the intakes of fiber and niacin and CVD mortality. Conversely, the intake of sugar and sodium, often implicated in CVD, was also observed to be lower in the CVD death group than in the control group, a finding that is in contrast with existing evidence [22]. This deviation might be indicative of the limitations of traditional analytical methods in the face of large datasets or potential deficiencies in the dietary data from the NHANES database, considering it only includes diet records from two days.

The performance of our CVD mortality prediction models in identifying at-risk individuals demonstrated a robust consistency across all metrics, as shown in Table 2. The random forest model showed the strongest performance among the health-based model. However, simpler models like logistic regression still delivered comparable results. This can be attributed to the tendency of random forest for overfitting with smaller datasets or higher outlier/noise influence [23].

The nutrition model exhibited lower accuracy than both the health and mixed models. This relative underperformance could be ascribed to the superior predictive power of health variables over nutrition variables in predicting CVD outcomes or to the insufficiency of nutrition data within the NHANES dataset. Nevertheless, the mixed model consistently surpassed both the health and nutrition models in performance across all algorithms, with the singular exception of ThunderSVM, which achieved its highest score within the health model. Rigdon et al. [10] also found best result when using random forest and dealing with dietary data.

Comparatively, these findings are consistent with those of previous research [9,10,24,25], which has demonstrated the utility of ML in health data analysis and the importance of considering both health and nutrition variables. Intriguingly, the study conducted by Rigdon and Basu [10] showed that incorporating nutrition data into the statistical model did not enhance its predictive discrimination or calibration. However, when this information was integrated into ML models, both these metrics improved. This bolsters the argument that ML models might be more efficient in exploiting the full potential of dietary information in predicting health outcomes, possibly by integrating more food-related variables and accounting for nonlinear and nonadditive relationships. Our results reinforce the benefits of leveraging both health and nutrition variables when predicting CVD outcomes, providing a more comprehensive representation of an individual’s health status.

The SHAP values shed light on the various health and nutrition variables that significantly influence the risk of death from CVD (Figure 2). Although direct comparison with previous studies presents challenges due to differences in variable selection, the majority of the risk factors listed in our chart align with findings from prior research. Age, systolic and diastolic blood pressure, and current smoker status have consistently been highlighted as significant variables in CVD classification or risk of death [8,24,26]. Glycated hemoglobin was also identified as a risk factor for CVD onset using ML methodologies [27]. While previous studies found cholesterol, education, and sex among their most relevant variables, these factors were present but did not rank among the top most important variables in our study (Appendix A). Intriguingly, our research highlights the often overlooked significance of lower eGFR and higher uric acid and lactate dehydrogenase levels, emphasizing the need for the comprehensive evaluation of other conditions such as gout or renal function in CVD risk assessment.

Regarding the impact of dietary variables, our study established that intakes of fiber, calcium, magnesium, polyunsaturated fatty acids, vitamin E, vitamin B6, protein, vitamin B2, potassium, and sodium were significant features of the ML model. Dihn et al. also reported calcium, fiber, and sodium asamong the most crucial variables in their ML analysis [27]. The positive association with vitamin B2 could be attributable to confounding factors, given that one of its main dietary sources is meat products, which have been previously reported to contribute to CVD [28]. Diets low or moderate in potassium, combined with higher potassium intake, are recommended to avert high blood pressure and decrease CVD mortality [29]. While we identified higher sodium intake as a crucial feature in our model, high potassium consumption was also associated with death from CVD. It is important to note, however, that the relationship between potassium and CVD mortality is multifactorial, influenced by factors such as age, kidney function (as indicated by eGFR), systolic blood pressure, and other nutritional components. These additional variables, which also ranked highly in our models, interact with potassium metabolism in intricate ways, potentially amplifying its effects on the cardiovascular system. In the mixed model, two variables emerged as key predictors for CVD-related mortality: fiber and vitamin E intake. Both of these nutrients have been extensively reported as preventing CVD [21,30].

The primary shortcoming of our research is that a majority of the nutritional data were collected through two 24 h dietary recalls. Moreover, many nutritional factors were omitted due to the issue of multicollinearity. This may have undermined our capacity to pinpoint significant factors and possibly generating some erroneous conclusions. Furthermore, discrepancies in age and sex distributions between the CVD-related deaths and the control group could have introduced biases. While an external dataset could have strengthened the validity of our findings, we did not employ one in this study. The adoption of larger datasets and recurring dietary assessments or food frequency questionnaires in the future could serve to alleviate this issue.

## 5. Conclusions

Not many studies have used dietary data for the prediction of CVD-related death. Our study, encompassing a broad spectrum of dietary variables, successfully pinpointed nutrients with a potentially significant role in CVD mortality prediction. Further investigation is crucial to determine whether these observed associations are causal. Our results emphasize the need for more extensive explorations into the use of machine learning for analyzing and leveraging complex dietary exposures in nutritional epidemiology. Artificial Intelligence (AI) methodologies present substantial potential for surpassing conventional and domain-specific methods in predictive accuracy, operational efficiency, cost-effectiveness, and user accessibility. Applying these techniques to large cohort studies or expansive data sources with recurring dietary recalls or food frequency questionnaires is expected to significantly amplify their overall value.

## Figures and Tables

**Figure 1 nutrients-15-03937-f001:**
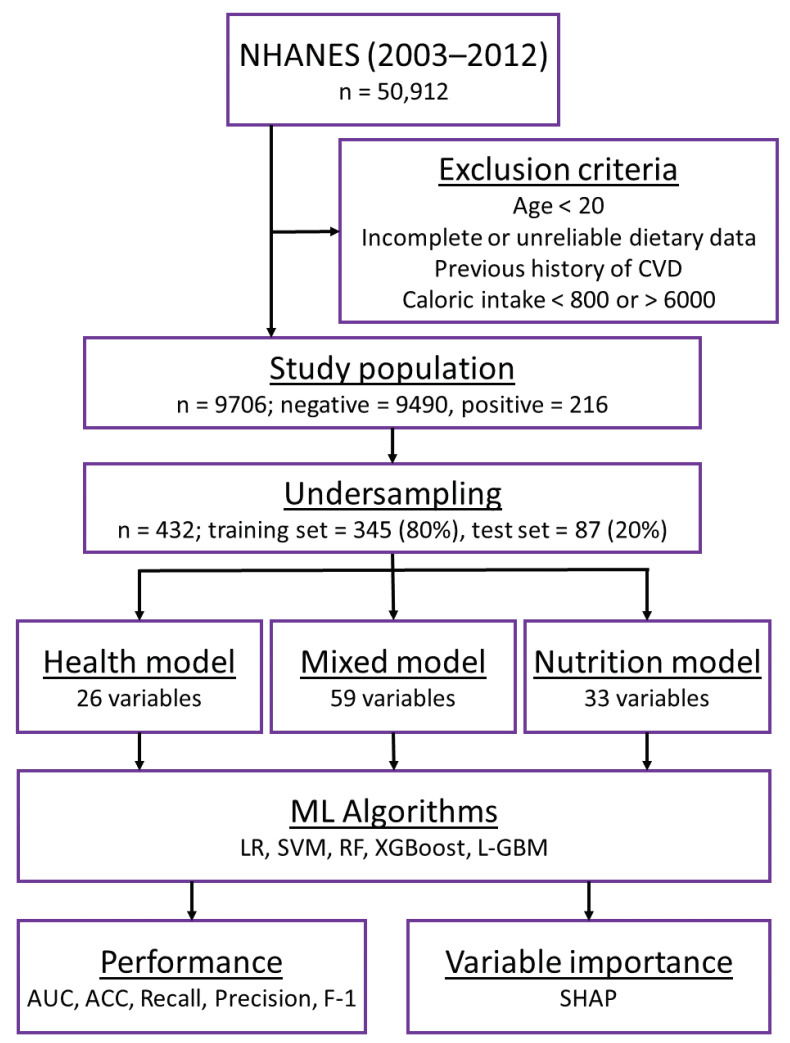
Flow chart of modeling. The sample size (*n*) and the positive/negative cases are shown inside the parentheses. NHANES, the National Health and Nutrition Examination Survey; LR, logistic regression; SVM, support vector machine; RF, random forest; XGBoost, eetreme gradient boosting; L-GBM, light gradient-boosting machine; SHAP, Shapley additive explanations.

**Figure 2 nutrients-15-03937-f002:**
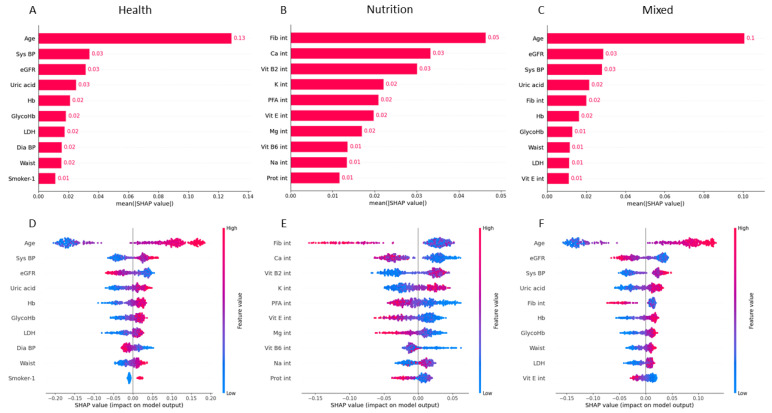
Feature importance for cardiovascular disease death based on SHAP values. On the top, the mean absolute SHAP values are depicted to illustrate global feature importance for health (**A**), nutrition (**B**), and mixed (**C**) models. On the bottom, the local explanation summary shows the relationship between a variable and death outcome for health (**D**), nutrition (**E**), and mixed (**F**) models. Positive SHAP values are indicative of positive correlation with CVD death, while negative SHAP values are indicative of negative correlation.

**Table 1 nutrients-15-03937-t001:** General characteristics of all model variables.

Variable	Control (*n* = 9490)	CVD Death (*n* = 216)	*p*-Value
**Health model variables**			
Age, years	43.0 [31.0, 57.0]	68.0 [60.0, 75.0]	<0.001
Sex			<0.001
Male	4600 (48.5)	148 (68.5)	
Female	4890 (51.5)	68 (31.5)	
Health Ins			<0.001
Yes	7525 (79.3)	199 (92.1)	
No	1965 (20.7)	17 (7.9)	
Education			<0.001
Less than 9th grade	2004 (21.1)	72 (33.3)	
High school grade	2049 (21.6)	65 (30.1)	
Some college, AA degree	2817 (29.7)	40 (18.5)	
College grad	2620 (27.6)	39 (18.1)	
Diabetic			<0.001
No	8646 (91.1)	161 (74.5)	
Yes	844 (8.9)	55 (25.5)	
Smoker			<0.001
Smoking	1782 (18.8)	60 (27.8)	
Quit	2207 (23.3)	77 (35.6)	
Never	5501 (58.0)	79 (36.6)	
Waist, cm	97.8 (15.3)	104.8 (14.9)	<0.001
Sys BP, mmHg	119.0 [110.0, 128.3]	131.0 [122.0, 144.0]	<0.001
GlycoHb, %	5.4 [5.2, 5.7]	5.7 [5.5, 6.1]	<0.001
K, mmol/L	3.9 [3.8, 4.1]	4.0 [3.9, 4.2]	<0.001
LDH, U/L	127.3 (21.3)	138.5 (20.5)	<0.001
Na, mmol/L	139.0 [138.0, 140.0]	139.7 [139.0, 140.5]	<0.001
Uric acid, mg/dL	5.3 [4.4, 6.1]	5.8 [5.3, 6.7]	<0.001
eGFR, mL/min/1.73 m^2^	103.3 (26.4)	81.4 (21.9)	<0.001
**Nutrition model variables**			
Ca int, mg	882.8 [630.5, 1214.4]	726.8 [516.9, 959.0]	<0.001
Cu int, mg	1.2 [1.0, 1.6]	1.1 [0.9, 1.4]	<0.001
Fib int, g	16.2 [11.7, 22.4]	13.9 [10.7, 18.0]	<0.001
Fol int, µg	389.0 [289.5, 528.0]	344.2 [270.2, 447.2]	<0.001
Mg int, mg	289.0 [225.0, 370.5]	260.0 [200.0, 312.6]	<0.001
Nia int, mg	24.1 [18.3, 31.5]	21.2 [16.5, 27.0]	<0.001
PFA int, g	16.3 [11.6, 22.4]	14.0 [9.7, 18.4]	<0.001
Prot int, g	80.9 [62.8, 102.9]	73.4 [58.8, 86.6]	<0.001
Na int, mg	3335.5 [2548.5, 4307.4]	2981.0 [2393.9, 3766.9]	<0.001
Sug int, g	107.8 [74.6, 150.5]	87.9 [61.3, 130.9]	<0.001
Vit B1 int, mg	1.6 [1.2, 2.1]	1.4 [1.1, 1.9]	<0.001
Vit B6 int, mg	1.9 [1.4, 2.6]	1.7 [1.2, 2.3]	<0.001
Vit E int, µg	6.9 [4.9, 9.9]	5.6 [4.2, 7.8]	<0.001

Numerical values are presented as the mean (standard deviation) if the distribution is approximately normal, or as the median [interquartile range] if the distribution is skewed. For categorical variables, percentages are given. Discrepancies in characteristics were assessed through the implementation of the unpaired Student’s *t*-test, Kruskal–Wallis test, or chi-squared test.

**Table 2 nutrients-15-03937-t002:** Performance metrics of machine learning algorithms for each model.

Model	Algorithms	ACC	AUC	Recall	Precision	*F*1-Score
Health	Logistic Regression	0.75	0.86	0.75	0.75	0.75
	Random Forest	**0.8**	0.87	**0.8**	**0.8**	**0.8**
	ThunderSVM	0.79	**0.88**	0.79	0.79	0.79
	XGBoost	0.78	0.86	0.78	0.78	0.78
	LightGBM	0.76	0.86	0.76	0.76	0.76
Nutrition	Logistic Regression	0.68	**0.7**	0.68	0.68	0.68
	Random Forest	**0.7**	**0.7**	**0.7**	**0.7**	**0.7**
	ThunderSVM	0.63	0.67	0.63	0.64	0.63
	XGBoost	0.66	0.69	0.66	0.66	0.66
	LightGBM	0.66	**0.7**	0.66	0.66	0.66
Mixed	Logistic Regression	0.76	0.86	0.76	0.76	0.76
	Random Forest	**0.82**	**0.88**	**0.82**	**0.82**	**0.82**
	ThunderSVM	0.77	0.87	0.77	0.77	0.77
	XGBoost	0.8	0.87	0.8	0.8	0.8
	LightGBM	0.79	0.87	0.79	0.79	0.79

ACC = balanced accuracy, AUC = area under the curve, and F1 score = harmonic mean of precision and recall.

## Data Availability

The data used in this study are publicly available online (https://wwwn.cdc.gov/nchs/nhanes/Default.aspx (accessed on 25 January 2023)).

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
