# Peer review of "Predicting Cardiovascular Disease Mortality: Leveraging Machine Learning for Comprehensive Assessment of Health and Nutrition Variables"

_nutrients, 2023, doi:10.3390/nu15183937_

Round 1
Reviewer 1 Report
The authors present a study to predict the CVD mortality by using health and nutritional variables. The idea is good, and the structure is well-organized. However, there are a couple of minor concerns to be addressed.
The authors mentioned that the dataset was then randomly partitioned into a training set with n = 345, while this was not consistent as Figure 1. Figure 1 does not show the training and test set.
How these important health and nutritional features were selected? Do the authors test other feature selection methods, such as logistic regression. Do they have the same results.
Do the authors use external dataset to verify these features?
NA.
Author Response
Comments 1: The authors mentioned that the dataset was then randomly partitioned into a training set with n = 345, while this was not consistent as Figure 1. Figure 1 does not show the training and test set.
Response 1: Thank you for pointing this out. We apologize for the oversight. We have now corrected Figure 1 to accurately represent the dataset partition as described in the text.
Figure 1 was updated. Page 4
Comments 2: How these important health and nutritional features were selected?
Response 2: Thank you for raising this question about the method to select health and nutritional features in our study.
For the nutritional features, our initial approach was to consider all nutritional variables from the NHANES datasets in their entirety. For health features, we primarily considered most of the biomarkers that were measured over the span of 10 years. However, upon analysis, we identified significant collinearity among these variables. Collinearity can make it challenging to discern the individual impact of each feature on the outcome. To address this, we utilized both the variance inflation factor (VIF) and correlation coefficients to identify and subsequently eliminate multicollinear features. We set a criterion of a VIF value of less than 10; variables exceeding this threshold were considered for removal. Beyond statistical considerations, the final selection of nutritional features was influenced by their established relevance in the literature concerning cardiovascular disease (CVD) mortality. For instance, total sugar intake, fiber intake, and total carbohydrate intake presented high collinearity. We decided to remove total carbohydrate intake based on the greater relevance of the other two variables to CVD and other metabolic syndrome-related diseases. Also, since one of the study's objectives is to assess the effect of adding nutritional variables to more well-known health variables, we decided to retain variables such as body mass index (BMI), cholesterol levels, hemoglobin, glycated hemoglobin, and others. Each of these has been underscored in the literature as being relevant to cardiovascular health.
In summary, our feature selection was a meticulous process that balanced statistical rigor with clinical relevance.
This selection procedure is explained in the “2.3. Data preprocessing and variable selection” section. Based on your comment we have added some further clarification for the nutritional variable selection method.
Text in manuscript revised. Page 5, lines 151-155
Comments 3: Do the authors test other feature selection methods, such as logistic regression. Do they have the same results.
Response 3: Thank you for your question. We did indeed used logistic regression during our preliminary analysis. Some results from this approach were consistent with those presented in the manuscript like fiber intake, age, and hemoglobin levels. On the other hand, certain variables like education level, race, and fish intake emerged as considerably more influential in the logistic regression analysis.
However, after careful consideration, we opted not to incorporate the logistic regression findings into this manuscript. We believe introducing a multitude of analytical techniques is out of the scope of our study and it might potentially overburden the reader, diverting attention from our main objective, which is to assess the usefulness of dietary data in predicting CVD death. Given the relative paucity of research harnessing nutritional data for this purpose, we believe that a focused approach will serve our readers best.
That said, for the sake of transparency and to accommodate your request, we have appended the logistic regression data below for your consideration. Feature importance is expressed as |z|.
Comments 4: Do the authors use external dataset to verify these features?
Response 4: We recognize the value that an external dataset could bring in validating our findings. However, it is difficult to find publicly available databases that align with our specific research objectives, possess sufficient nutritional data, and are compatible with the data presented in our study. We have modified the limitations of our study in the last paragraph of the discussion section to address your comment.
Text in manuscript revised. Page 11, lines 374-376
In conclusion, I wish to express my gratitude to the reviewers for their insightful comments and suggestions. They have significantly contributed to improving the quality of the manuscript. I hope that the revisions addressed all concerns, and I look forward to your further feedback.
Sincerely,
Agustin Martin-Morales
National Institutes of Biomedical Innovation, Health and Nutrition

Reviewer 2 Report
Given its retrospective nature, this study seeks to predict CVD mortality and ascertain risk factors within 10 years following the collection of individual measurements. It's imperative to establish this timeframe in advance, as subsequent time intervals might compromise the model's precision. Furthermore, individual records may have undergone changes during the 10-year period. The study's reliance on single survey data for explanatory variables, even though NHANES conducts biennial surveys, restricts its ability to depict the trajectory of risk factors and disease progression. A survival analysis or longitudinal design is more likely to accurately capture the timely relationships between risk factors and disease advancement.
This analyzed cohort comprised a relatively small sample size, potentially impacting the model's performance due to the influence of the training data. Numerous participants with missing data were excluded from the analysis, which leads to a concern of sample representative.
To address missing data, it is advisable for the authors to utilize a multiple imputation approach, excluding only samples with significant missing data. This method is particularly recommended when the missing values constitute less than 30% of the other variables, aiming to ameliorate issues related to sample integrity.
Individuals who experienced CVD-related deaths were predominantly elderly, with a median age of 68 years, in contrast to the control group's median age of 43 years, and a higher proportion of them were male. Due to these discrepancies, the design of the control group might not be optimal. Notably, Table 1 underscores substantial differences in all predictors between the CVD and control groups. These disparities might be influenced by the age distribution, potentially introducing biases.
The study underscores the significance of a comprehensive health evaluation and dietary intake assessment in predicting CVD-related mortality. However, the mixed model only exhibits a marginal 0.02 improvement in metrics compared to the health model within the optimal model. Feature importance analysis using SHAP also highlights health-related features as the primary contributors to the models. Considering the aspect of method simplicity, the health model might be more appropriate for real-world applications.
Table 1 provides an overview of variables for both the whole control and CVD-death groups. It's essential to present the characteristics of variables for the control group in the context of the under-sampling model as well.
Medical treatment emerges as a significant confounding factor for CVD patient mortality. Regrettably, the study overlooks an investigation into individual medical interventions preceding death.
Author Response
Comments 1: Given its retrospective nature, this study seeks to predict CVD mortality and ascertain risk factors within 10 years following the collection of individual measurements. It's imperative to establish this timeframe in advance, as subsequent time intervals might compromise the model's precision. Furthermore, individual records may have undergone changes during the 10-year period. The study's reliance on single survey data for explanatory variables, even though NHANES conducts biennial surveys, restricts its ability to depict the trajectory of risk factors and disease progression. A survival analysis or longitudinal design is more likely to accurately capture the timely relationships between risk factors and disease advancement.
Response 1: The primary aim of our study was to identify risk factors, both health and nutritional, associated with CVD mortality within a defined timeframe (10 years post collection of individual measurements) using the NHANES data. While survival or longitudinal analyses indeed provide a more detailed understanding of disease progression over time, the inclusion of such analysis would introduce a different dimension to our research, potentially diverting from our main objective. Given the complexity and vast scope of such analyses, they might be more appropriate for a separate, dedicated study, rather than as an adjunct to our current research.
Our decision to utilize data form the NHANES from 2003 to 2012 was made with care. Several variables' measures changed between the years, making it challenging to harmoniously integrate data after 2012 without risking data inconsistency. Moreover, as the reviewer rightly pointed out, ensuring the precision of our model was of utmost importance. As the data grew more recent, we observed fewer cases of death from CVD. This discrepancy created a vast difference between the CVD group and the control group, which could have biased our results. Our research's discussion section delves deep into the associations and deviations we identified from our data, and we believe these findings provide valuable insights into CVD risk factors within our chosen time span.
Comments 2: This analyzed cohort comprised a relatively small sample size, potentially impacting the model's performance due to the influence of the training data. Numerous participants with missing data were excluded from the analysis, which leads to a concern of sample representative.
To address missing data, it is advisable for the authors to utilize a multiple imputation approach, excluding only samples with significant missing data. This method is particularly recommended when the missing values constitute less than 30% of the other variables, aiming to ameliorate issues related to sample integrity.
Response 2: We understand your concerns about sample size and missing data. Although our cohort size was relatively small, the consistency of our CVD mortality prediction models, as indicated in Table 2, suggests the validity of our findings even with the current sample size. Our results, consistent with existing literature, support the importance of our conclusions despite the sample limitations.
On the matter of imputation, we genuinely apologize for the oversight in our materials and methods section. You are correct in your assessment. We did indeed employ a multiple imputation approach to address the missing data, aligning with the recommendations you provided. This was unintentionally omitted in our initial submission, and we have now made the necessary modifications to the manuscript to reflect this (see “2.3. Data preprocessing and variable selection” section. We concur that this method is essential for maintaining the integrity of our sample, especially when missing values are below the 30% threshold.
Text in manuscript revised. Page 3, lines 122-124
Comments 3: Individuals who experienced CVD-related deaths were predominantly elderly, with a median age of 68 years, in contrast to the control group's median age of 43 years, and a higher proportion of them were male. Due to these discrepancies, the design of the control group might not be optimal. Notably, Table 1 underscores substantial differences in all predictors between the CVD and control groups. These disparities might be influenced by the age distribution, potentially introducing biases.
Response 3: Thank you for your observations regarding the age and gender disparities between the CVD-related deaths and the control group. We acknowledge these differences and considered them thoroughly during our analysis. The nature of CVD-related deaths means that they are more prevalent among the elderly, which explains the median age difference between our groups. It is an inherent characteristic when studying CVD death, and hence the expected age distribution in our sample reflects the real-world scenario. Our choice of control group was made with consideration to achieve a balance between general population representation and minimizing potential confounders. We have added your comment in our last paragraph of our discussion section as “Furthermore, discrepancies in age and gender distributions between the CVD-related deaths and the control group may introduce potential biases.”
Text in manuscript revised. Page 11, line 373
Comments 4: The study underscores the significance of a comprehensive health evaluation and dietary intake assessment in predicting CVD-related mortality. However, the mixed model only exhibits a marginal 0.02 improvement in metrics compared to the health model within the optimal model. Feature importance analysis using SHAP also highlights health-related features as the primary contributors to the models. Considering the aspect of method simplicity, the health model might be more appropriate for real-world applications.
Response 4: Thank you for your comment. You rightly pointed out that the mixed model's improvement in predictive metrics is marginal compared to the health model. However, the importance of this model lies in its comprehensive representation of an individual's health status, integrating both health and nutritional variables. As stated in our discussion, “Our results reinforce the benefits of leveraging both health and nutrition variables when predicting CVD outcomes.” The inclusion of dietary variables in the mixed model not only provides additional insights but also lays the groundwork for further research into the role of specific nutrients in CVD mortality prediction.
Additionally, our SHAP values analysis showcases the impact of specific dietary variables on the model's decision-making, suggesting that even though health-related features are the primary contributors, dietary variables also play a non-trivial role in the predictive power of the mixed model. As we emphasized in the discussion, “In the mixed model, two variables emerged as key predictors for CVD-related mortality: fiber and vitamin E intake,” further demonstrating the benefits of considering dietary variables in conjunction with health metrics.
Lastly, while we recognize the argument for method simplicity in real-world applications, the mixed model's added complexity is justified by its holistic approach, which might be crucial in certain medical or research contexts where a comprehensive health evaluation is desired.
Comments 5: Table 1 provides an overview of variables for both the whole control and CVD-death groups. It's essential to present the characteristics of variables for the control group in the context of the under-sampling model as well.
Response 5: We agree presenting undersampling data is important. We chose not to include it in the manuscript to avoid overwhelming the reader, especially given that the data was consistent with what has already been presented. We have appended the requested table below for your consideration. We also have modified the result section acknowledging that the undersampling data was consistent with the one presented in Table 1.
Text in manuscript revised. Page 6, line 209
Comments 6: Medical treatment emerges as a significant confounding factor for CVD patient mortality. Regrettably, the study overlooks an investigation into individual medical interventions preceding death.
Response 6: Thank you for raising the valid concern regarding the potential confounding effects of medical treatment on CVD patient mortality. We recognize the significance of medical interventions and the implications they might have on our analysis.
However, we would like to clarify the decision to utilize data from the NHANES conducted between 2003 and 2012. While medical treatment is indeed a crucial factor to consider, the inclusion of patients starting CVD medication after 2012 might introduce bias. This is because our primary dataset includes participants from 2003 to 2012. Patients taking medication after 2012 would escape our analysis, and individuals commencing their medication post-2012 could have been exposed to newer therapeutic approaches or medications which were not prevalent or available during the 2003-2012 period. This discrepancy in treatment regimens between groups could indeed skew our results.
Nonetheless, we acknowledge the importance of the reviewer's suggestion. In future research, we will aim to integrate a more comprehensive evaluation of medical treatments, specifically considering their onset date, to ascertain their potential confounding influence on CVD mortality.
In conclusion, I wish to express my gratitude to the reviewers for their insightful comments and suggestions. They have significantly contributed to improving the quality of the manuscript. I hope that the revisions addressed all concerns, and I look forward to your further feedback.
Sincerely,

Reviewer 3 Report
The study comprehensively assess the health and nutrition variables to identify risk factors associated with CVD mortality. This study contributes to the growing body of research on the relationship between diet, lifestyle, and CVD risk. Its findings underscore the potential importance of dietary factors in CVD prevention and highlight the need for further research in this area. Controversial results regarding intakes of vitamin B2 and potassium were discussed and the Aurthors tried to explain them. The manuscript is well written, according to the journal's request. Below I present a few of my comments:
1. In Figure with Flow chart of modelling the exclusion criterion “800 > Kcal>6000” is unclear, please correct.
2. Under Table 1 please unify a description to be compatible with the method of presentation e.g. please change type of parentheses: mean (standard deviation) if the distribution is approximately normal, or as the median [interquartile range] (“[interquartile range]” instead of “(interquartile range)”)
3. While comparing Panel A Figure 2 with a panel D it seems that some inconsistency occur. Shouldn’t be in some variables (Dia BP, eGFR) negative correlation inserted in panel A?. why the coefficient for EGFR and Dia BP is positive? You put only “the strength” of importance or also direction? If only strength – I think “plus” in each coefficient is misleading.
4. In the sentence “The performance of our CVD mortality prediction models in identifying at-risk individuals demonstrated a robust consistency across all metrics 2. – should be reference to “Table 2”
5. “Regarding the impact of dietary variables, our study established that increased intakes of fiber, calcium, magnesium, polyunsaturated fatty acids, vitamin E, vitamin B6, and protein, and reduced intakes of vitamin B2, potassium, and sodium were significant features of the ML model” – are in such sentence the words “increased” and “decresed” needed? In the context of association with CVD mortality – to show direction yes, but here only “significance” is reported.
Author Response
Comments 1: In Figure with Flow chart of modelling the exclusion criterion “800 > Kcal>6000” is unclear, please correct.
Response 1: Thank you for pointing this out. We have now corrected Figure 1 to accurately represent the exclusion based on caloric intake.
Figure 1 was updated. Page 4
Comments 2: Under Table 1 please unify a description to be compatible with the method of presentation e.g. please change type of parentheses: mean (standard deviation) if the distribution is approximately normal, or as the median [interquartile range] (“[interquartile range]” instead of “(interquartile range)”)
Response 2: We appreciate your suggestion regarding the consistent presentation of data. We modified the table to reflect the standardized format: mean (standard deviation) for normal distributions and median [interquartile range] for non-normal distributions.
Table 1 footnote was updated. Page 7
Comments 3: While comparing Panel A Figure 2 with a panel D it seems that some inconsistency occur. Shouldn’t be in some variables (Dia BP, eGFR) negative correlation inserted in panel A?. why the coefficient for EGFR and Dia BP is positive? You put only “the strength” of importance or also direction? If only strength – I think “plus” in each coefficient is misleading.
Response 3: We appreciate your observation regarding the depiction of Dia BP and eGFR in Panel A. Upon revisiting the data, we concur with your assessment. The "plus" sign, it was initially incorporated to signify strength using absolute values. However, considering your feedback, we recognize the potential for ambiguity. To ensure clarity and precision in our presentation, the figure has been modified.
Figure 2 (A, B and C) was updated, + sign was removed from absolute values. Page 9 (supplementary data will be modified accordingly)
Comments 4: In the sentence “The performance of our CVD mortality prediction models in identifying at-risk individuals demonstrated a robust consistency across all metrics 2. – should be reference to “Table 2”
Response 3: We apologise for this mistake. The sentence in question has been revised as: “The performance of our CVD mortality prediction models in identifying at-risk individuals demonstrated a robust consistency across all metrics, as shown in Table 2.”
Text in manuscript revised. Page 10, line 314
Comments 5: “Regarding the impact of dietary variables, our study established that increased intakes of fiber, calcium, magnesium, polyunsaturated fatty acids, vitamin E, vitamin B6, and protein, and reduced intakes of vitamin B2, potassium, and sodium were significant features of the ML model” – are in such sentence the words “increased” and “decreased” needed? In the context of association with CVD mortality – to show direction yes, but here only “significance” is reported.
Response 3: Thank you for pointing out the potential redundancy in our description of the dietary variables. Upon reflection, we concur that when discussing mere significance, without specifically addressing the direction of association, the terms "increased" and "decreased" may be superfluous. We appreciate this insight.
The sentence was revised as: "Regarding the impact of dietary variables, our study established that intakes of fiber, calcium, magnesium, polyunsaturated fatty acids, vitamin E, vitamin B6, protein, vitamin B2, potassium, and sodium were significant features of the ML model."
Text in manuscript revised. Page 10, line 352-354
Once again, we appreciate your insightful comments. We hope that our revisions will address your concerns, and we look forward to any further feedback.
Sincerely,
Agustin Martin-Morales
National Institutes of Biomedical Innovation, Health and Nutrition
Round 2
Reviewer 2 Report
The study might collect individual interview data at various points over the ten-year period. It is crucial to account for the time elapsed between exposure and the occurrence of the event (i.e., death) for a predictive study. For instance, it is not reasonable to indicate the same level of risk of mortality to a person who dies one year after exposure compared to someone who passes away ten years after exposure. Regrettably, the study did not address the issue of time intervals, which could potentially reduce the depth of scientific insight.